# Cardiovascular Risk Perception and Knowledge among Italian Women: Lessons from IGENDA Protocol

**DOI:** 10.3390/jcm11061695

**Published:** 2022-03-18

**Authors:** Silvia Maffei, Antonella Meloni, Martino Deidda, Susanna Sciomer, Lucia Cugusi, Christian Cadeddu, Sabina Gallina, Michela Franchini, Giovanni Scambia, Anna Vittoria Mattioli, Nicola Surico, Giuseppe Mercuro

**Affiliations:** 1Cardiovascular and Gynaecological Endocrinology Unit, Fondazione G Monasterio CNR-Regione Toscana, 56124 Pisa, Italy; 2Department of Radiology, Fondazione G Monasterio CNR-Regione Toscana, 56124 Pisa, Italy; antonella.meloni@ftgm.it; 3Department of Medical Sciences and Public Health, University of Cagliari, 09042 Cagliari, Italy; martino.deidda@tiscali.it (M.D.); cadedduc@unica.it (C.C.); giuseppemercuro@gmail.com (G.M.); 4Department of Cardiovascular, Respiratory Nephrological, Anesthesiological and Geriatric Sciences, Sapienza University, 00186 Roma, Italy; susanna.sciomer@uniroma1.it; 5Department of Biomedical Sciences, University of Sassari, 07100 Sassari, Italy; lucia.cugusi@uniss.it; 6Department of Neuroscience, Imaging and Clinical Sciences, University of Chieti-Pescara, 66100 Chieti, Italy; sgallina@unich.it; 7Epidemiology and Health Research Lab, Institute of Clinical Physiology, National Research Council, 56124 Pisa, Italy; michela.franchini@ifc.cnr.it; 8Gynecologic Oncology, Department of Woman and Child Health and Public Health, Fondazione Policlinico Universitario A. Gemelli, IRCCS, 00168 Roma, Italy; giovanni.scambia@policlinicogemelli.it; 9Surgical, Medical and Dental Department of Morphological Sciences Related to Transplant, Oncology and Regenerative Medicine, University of Modena and Reggio Emilia, 41124 Modena, Italy; vittoria@unimore.it; 10Department of Translational Medicine, Università Piemonte Orientale, 28100 Novara, Italy; nicola.surico@med.unipmn.it

**Keywords:** awareness, cardiovascular disease, cardiovascular risk factors, knowledge, perception, women

## Abstract

A multicenter, cross-sectional observational study (Italian GENder Differences in Awareness of Cardiovascular risk, IGENDA study) was carried out to evaluate the perception and knowledge of cardiovascular risk among Italian women. An anonymous questionnaire was completed by 4454 women (44.3 ± 14.1 years). The 70% of respondents correctly identified cardiovascular disease (CVD) as the leading cause of death. More than half of respondents quoted cancer as the greatest current and future health problem of women of same age. Sixty percent of interviewed women considered CVD as an almost exclusively male condition. Although respondents showed a good knowledge of the major cardiovascular risk factors, the presence of cardiovascular risk factors was not associated with higher odds of identifying CVD as the biggest cause of death. Less than 10% of respondents perceived themselves as being at high CVD risk, and the increased CVD risk perception was associated with ageing, higher frequency of cardiovascular risk factors and disease, and a poorer self-rated health status. The findings of this study highlight the low perception of cardiovascular risk in Italian women and suggest an urgent need to enhance knowledge and perception of CVD risk in women as a real health problem and not just as a as a life-threatening threat.

## 1. Introduction

The cardiovascular (CV) risk involves anthropometric, metabolic, biological, and behavioral factors, which, in turn, interact with sex and are modulated by aging [1,2,3]. CV risk is not uniformly distributed across the world, as it is also influenced by genetics, lifestyle, environment, and the culture of each population. Countries, therefore, can be divided into two main groups, based on the high or low risk of CV disease (CVD) of their populations. In Europe, Italy is a low-CV-risk country [4], although CVD is also in this country the leading cause of disability and death.

Objective CV risk assessment (e.g., Framingham score) does not include the individual knowledge of CV risk or CVD prevalence [5,6,7]. This lack has a strong impact on the problem and its correction, because to adopt a healthy lifestyle, a correct perception of personal risk is necessary: its deficiency is to be considered almost as an additional risk factor [8]. The problem is even more significant among women, due to the wrong and settled belief that CVD is primarily, if not exclusively, a male disease. In turn, this misunderstanding is generated by the differences in the clinical presentation of CVD in the two sexes and by the increased prevalence of CVD in women of advanced postmenopausal age [8,9]. Moreover, several female-specific risk factors for CVD, such as polycystic ovary syndrome, premature ovarian failure, hypertensive disorders of pregnancy, gestational diabetes, and preterm birth, are underappreciated clinically and not receive adequate attention in CVD risk prediction [10,11,12].

A lack of knowledge of CV risk factors (CVRF) among women has been demonstrated by the majority of published studies [13,14,15,16,17,18,19,20,21]. The knowledge that CVD is the leading cause of death proved to be suboptimal among Australian and US women [19,21,22]. A nationwide survey conducted by the Women’s Heart Alliance in 2014 to determine barriers and opportunities for women and physicians with regard to CVD showed that almost half of 1011 interviewed women did not recognize CVD as the number one killer of women in the US [23]. Conversely, in a survey conducted in Austria, 75.3% of women correctly defined CVD as the leading cause of death [18].

Instead, with regard to the assessment of the impact of the CVD on their personal health status, several studies revealed a wrong perception even amongst women with CVRF [14,22,24]. For example, Mosca et al., in their survey of 2010, found that only 16% of women recognized the CVD as their biggest health problem, while 46% perceived cancer as the leading cause of illness in the female gender and the greatest threat to their future health [22].

It should be recognized that 90% of the published studies investigated US and Australian women [14,19,21,22,24], while the European situation in terms of knowledge and perception of CVD remains almost totally unexplored.

To the best of our knowledge, only one population-based study was conducted in Italy [25], which showed that only 26.5% of respondents correctly identified the main CVRF, with the exception for smoking (89.4%) and high cholesterol level (74.7%). Unfortunately, this study involved only 830 women recruited at random in five public schools (mothers of children aged 3 to 18 years), who lived in a single, limited urban area (Naples and Salerno). Therefore, this cohort cannot be considered representative of the entire Italian female population.

On the basis of these inadequate premises, we conducted a national CV health survey on a large sample of 5000 women attending one of the 80 outpatient gynecological centers throughout the Italian Society of Obstetrics and Gynecology (SIGO) network [26]. Indeed, all the project phases were conducted in strict collaboration with the Italian Society of Cardiology (SIC) and SIGO. In this representative female population, we investigated the degree of knowledge of CVD and CVRF and the perception of CV health status.

## 2. Materials and Methods

### 2.1. Study Design and Participants

The Italian GENder Differences in Awareness of CV risk (IGENDA) is a cross-sectional, observational, multicenter study whose protocol has already been described [26]. As part of this study, an anonymous, self-administered questionnaire was distributed to 5000 consecutive Italian women aged 18 to 70 years, who attended one of the 80 outpatient gynecological centers through the SIGO network. Exclusion criteria were a diagnosis of malignancy in the last 5 years and/or an ongoing chemotherapy treatment.

The study was conducted in accordance with the Declaration of Helsinki [27] and the GCP guidelines of the European Commission. Ethical approval was obtained from the Ethical Committee of Pisa (Comitato Etico Area Vasta Nord Ovest) on 13 March 2014 (Protocol Number 17857). All participants gave their signed informed consent to the project and received a brief description of the study’s aims and how to complete the questionnaire.

### 2.2. Structure of the Questionnaire

The questionnaire for the participants included the following three main parts.

Part 1, baseline assessment, collected general information on survey participants as well as their knowledge of CVD as the greatest health problem or risk.

Part 2, on CVDs, investigated the knowledge of traditional CVRF: women were asked if they were aware on the fact that a given condition/behavior represented a CVRF.

Part 3, on individual health status and perceived CVD risk. Participants were asked on their individual CVD history, such as pre-existing coronary artery disease, myocardial infarction, or stroke. The perception of subjective health status and personal CVD risk profile was assessed by a Self-Rating of Health (SRH), using the single-items “How would you rate your current health status on a scale from 0 to 10?” and “Which is your personal risk of developing a CVD in the future on a scale from 0 (very low) to 10 (very high)?”. Ratings for SRH were poor <4, acceptable 5–7, and good 8–10. Similarly, ratings for perceived risk were low <4, intermediate 5–7, and high 8–10.

The completion of the questionnaire required at least 20 min.

Data on the educational level of the participants were included in a general health questionnaire compiled by their physicians.

### 2.3. Statistical Analysis

Data analyses were conducted with the IBM SPSS Statistics 20 statistical package. Continuous variables were described as mean ± standard deviation (SD) and categorical variables as frequencies and percentages. The normality of distribution of the parameters was assessed by using the Kolmogorov–Smirnov test. For continuous values with normal distribution, comparisons between groups were made by independent-samples *t*-test (for two groups) or one-way ANOVA (for more than two groups). Wilcoxon’s signed rank test and Kruskal–Wallis test were applied for continuous values with non-normal distribution. The Bonferroni correction test was used in all pairwise comparisons. The χ^2^ testing was performed for non-continuous variables. The odds ratios (ORs) and 95% confidence intervals (CIs) were calculated using logistic regression analysis.

A two-tailed probability *p* < 0.05 was considered statistically significant.

## 3. Results

### 3.1. Characteristics of Respondents

A total of 4454 respondents filled out the questionnaire. Their demographic and health characteristics are summarized in Table 1. The mean age was 44.3 ± 14.1 years, with 62% of participants over 40 years old. Three quarters of the women had a medium to high education level.

The 58.2% of women had one or more CVRF; in detail, 44.5% had a single CVRF, 25.4% reported two CVRF, and the remaining 30.1% reported three or more CVRF. Among the CVRF, the following frequencies were detected: menopause 27.7%, obesity/overweight 19.2%, smoking habit 17.9%, hypertension 17.5%, family CVD history 17.1%, dyslipidemia 13.0%, and diabetes mellitus 6.7%.

The 2.8% of respondents reported to suffer from a diagnosed CVD. Women with CVD were significantly older than those without CVD (54.9 ± 14.9 vs. 42.1 ± 13.4; *p* < 0.0001).

### 3.2. Knowledge of CVD

The 69.8% of women correctly identified CVD as the leading cause of death in the Italian population (Figure 1a). Those who gave the correct answer were older (45.1 ± 14.1 vs. 42.2 ± 14.1 years; *p* < 0.0001) and had the highest level of education (university vs. secondary school 74.0% vs. 65.8%; *p* < 0.0001, and university vs. high school, 74.0% vs. 68.5%; *p* = 0.006). The percentage of women who answered this question correctly was similar in subjects with and without CVD (76.0% vs. 69.0%; *p* = 0.134). Furthermore, this percentage was significantly higher among women with at least one CVRF compared to those who reported none (70.5% vs. 65.7%; *p* = 0.005).

The 60.8% of women believed that CVD is an almost exclusively male condition (Figure 1b).

In the multiple choice questions, more than half of women identified cancer as the biggest health problem for people of the same age and gender (74%) and as the greatest danger to their health in the future (64%) (Figure 2). Only 20.7% and 24.6% of women, respectively, selected CVD in the two aforementioned questions. Noteworthily, these women most often suffered from CVD (4.6% vs. 2.2%, *p* = 0.002; OR 2.1, 95% CI: 1.4–3.4; *p* = 0.001) and were older (46.6 ± 14.3 vs. 43.3 ± 14.1 years; *p* < 0.0001).

### 3.3. Knowledge of the Main CVRF

Table 2 shows respondents’ degree of knowledge of CVRF and how age and level of education, as well as the presence of CVRF or CVD, influences this knowledge. Greater recognition of all CVRF was found in progressively older age groups and among women with the highest level of education. Women affected by CVRF correctly identified only family history of CVD, high cholesterol, and menopause as CVRF. The presence of CVD did not correlate with a higher level of knowledge of the individual CVRF.

### 3.4. Self-Rating of Health

The characteristics of the three groups based on SRH status are presented in Table 3. The 10.4% of women rated their health as bad, 47.6% as acceptable, and 42.0% as good. Age was significantly higher in the poor SRH group than in both the acceptable and good SHR groups (*p* < 0.0001), as well as in the acceptable SRH group in comparison with good SRH (*p* = 0.003). Women who reported good SRH had significantly lower prevalence of both CVRF and CVD than respondents with acceptable and poor SRH (*p* < 0.0001; Table 3).

### 3.5. Perception of CVD Risk

The 44.5% of the women considered themselves to be at low CVD risk and 46.7% at intermediate risk, although the prevalence rates of CVRF in the two groups were 50.2% and 64.6%, respectively. Only the remaining 8.8% of the respondents reported to perceive a high CVD risk, commensurated with a 70.0% CVRF prevalence (Table 4). A direct correlation between perceived risk increase and age increase was detected: the group with a low perceived CVD risk was significantly younger than the groups of women with both intermediate and high perceived CVD risk (*p* < 0.0001). Finally, a good SRH status was associated with a low perceived CVD risk (Table 4).

## 4. Discussion

This national survey documented several important findings. In Italy, women’s knowledge of CVD as the main cause of death is higher (69.8%) than those reported by the US (56%) [28] and Australian (32%) [19] women and comparable to that found among Austrian women (75.3%) [18]. In Italy, the knowledge of CVD as the main cause of death was more associated with older age than with the presence of CVD, thereby suggesting that such consciousness may be influenced by life experiences (e.g., number of CVD deaths among friends and relatives).

On the other hand, most of the interviewed women, although correctly identifying the CVD as the main cause of death in Italy, considered themselves at risk of oncological diseases rather than CVD. This misperception was even more evident in younger women, in whom a further misunderstanding of CV risk could be due to the poor attention to their health status and lack of life experience of CVD. The apparent contradiction between the knowledge of CVD as a leading cause of mortality and the expectation of cancer in future life could be largely explained by the erroneous but widespread opinion that CVD is more common in males. Consistently, our data showed that over 60% of the respondents considered CVD as an almost exclusively male condition.

Italian women showed in our study a good knowledge of the main CVRF, unlike the US and Australian female populations [13,14,15,16,17,19,20]. Actually, in contrast to our data, the only previous population study conducted in Italy also showed a very poor knowledge of the main CVRF among women (26.5%) [25]. The discrepancy in the results could be largely explained by the small sample interviewed in that study, the survey area limited to two cities in southern Italy, and the low percentage of people over 60 years old. Moreover, the study by Tedesco was performed years ago and before the national campaign of information on CV risk [25]. Being overweight was the most commonly recognized CVRF by women with no differences between age groups. It is reasonable to assume that the widespread attention to the aesthetic aspect typical of our era contributes to this awareness. Conversely, menopause was the less frequently identified CVRF. This finding corresponds and could be explained by the underestimation of menopause as an individual CVRF also by the scientific community. Furthermore, younger women were less aware of this relevant problem than the postmenopausal respondents. Diabetes was one of the most underestimated CVRF in our survey, despite the recognized importance of diabetes as a CVD equivalent [1]. These data are in accordance with previous surveys [16,18], where diabetes was the less known CVRF, identified by less than half of the women surveyed. The knowledge of CVRF was strongly influenced by age and educational level, confirming the positive effect of schooling, in accordance with data reported by other studies [25,29]. Inexplicably, the presence of a known CVD did not improve the knowledge of the CVRF among women. This feature proved controversial in previous surveys. While some studies showed that women with CVRF or CVD have greater knowledge of CV risk [14,15,30], Hoare et al. [19] revealed only negligible differences in the knowledge of clinical CVRF among women with CVD or diabetes compared to healthy women. It has consistently been shown that women with diabetes and hypertension did not identify these conditions as CVRF [17] and that less than 50% of them had an exact knowledge of the normal cholesterol, blood pressure, or blood glucose [14,21,31]. Besides menopause, the knowledge of women-specific CVRF was not evaluated, but we are planning to explore this issue in our population. Two studies conducted in the USA showed that women with pre-eclampsia or gestational diabetes mellitus were unfamiliar with the relationship between these conditions and increased future CVD risk [32,33].

In this study, we assessed the female perception of health status and personal CVD risk profile by means of the SRH status assessment. This is a subjective reflection of one’s general personal health condition, called “perceived” or “subjective” health, that integrates biological, mental, social, and functional aspects of a person, including individual and cultural beliefs and health behaviors [34]. As expected and in agreement with previous studies [35,36], we found that age was negatively correlated with the SRH status (younger women feel better than middle-aged and older women) and that a worse SRH status was associated with a more frequent reporting of CVRF and CVD [37].

In spite of these findings and their good knowledge of CVRF, our interviewees did not translate this information into a correct perception of their personal risk of developing CVD: although this insight would increase with age, higher CVRF and CVD reporting, and poorer SRH status, less than 10% of all women perceived the high possibility of CVD in their future life. Certainly, this lack of awareness is more concerning in women who reported having at least one CVRF, as more than one-third of them perceived their future CVD risk as low. These results confirm previous investigations. A Canadian study found that a significant percentage of participants were unaware of their CVD risk status [16]. Furthermore, a survey conducted on a smaller sample of elderly women (>70 years) confirmed that those with CVRF or CVD had a misperception of their health status, so that only 24.5% of the high-risk group—based on BMI, high blood pressure, cholesterol levels, and smoking habits—failed to recognize their actual risk of CVD [31]. These findings highlight a misperception of present and future health status in relation to the actual presence of CVRF, a mechanism for removing one’s own risk of disease that has cultural and social roots. Women continue to be the reference for the whole family and are the family caregiver. They see and recognize the risk of illness in their family members but do not recognize their own risk, an attitude that can be traced back to the theory known in health psychology as “comparative optimism”. The “comparative optimism” is the belief that negative events are more likely to happen to others, while positive events are more likely to happen to themselves [38]. It has been shown that people are likely to retrieve the information on the general perception of risk through social influence; however, when considering self-reported risk, positive prejudices are likely much more influential, with the consequence that people underestimate their personal risk [39]. 

## 5. Conclusions

Our data highlight the need to strengthen the knowledge of CVD, as well as CVRF, as a real health problem for both the sexes, removing the still widespread misunderstanding of a “male-limited” condition. For this purpose, a global campaign to improve knowledge and perception of CVD is of paramount importance. This campaign should be conducted within the health system, during patient visits, and by organizing specific seminars or conferences directed to the general population, and, above all, through various digital communication channels, including mass media (TV and radio advertisements and articles on newspapers and magazines), social media (blogs, micro-blogs, wikis, social networking sites, photo-sharing sites, instant messaging, video-sharing sites, and podcasts), and text messages or phone calls. The digital communication campaign can reach audiences in a low-cost, impactful, and effective way. Younger women should be the main focus of this educational project. Indeed, this is the key to providing a more prevention-oriented approach, a powerful and decisive contribution to reducing CVD risk among women.

## Figures and Tables

**Figure 1 jcm-11-01695-f001:**
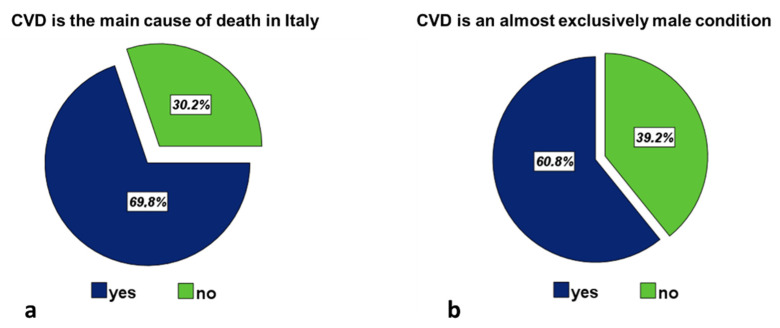
(**a**) Percentages of women who identified/did not identify CVD as the leading cause of death in Italy. (**b**) Percentage distribution of women who stated or excluded that CVD is an almost exclusively male condition.

**Figure 2 jcm-11-01695-f002:**
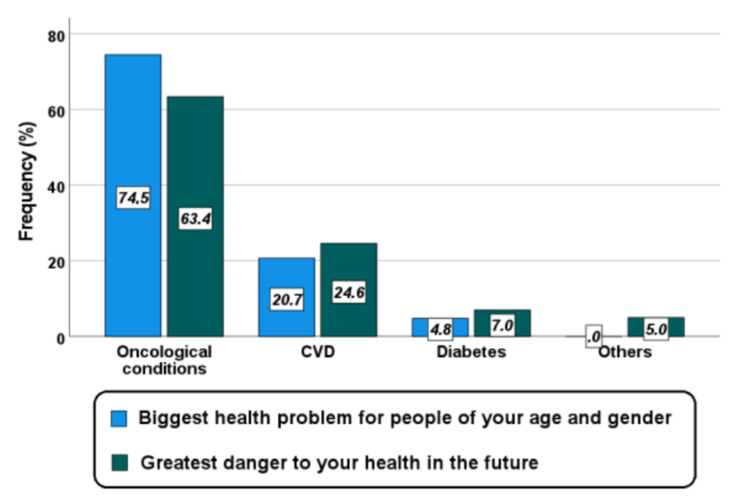
Percent response to the multiple choice questions “What is the biggest health problem for people of your age and gender and what is the greatest danger to your health in the future?”.

**Table 1 jcm-11-01695-t001:** Demographics and health characteristics of respondents.

Age Group	*n* (%)
<30 years	820 (18.4)
30–39 years	851 (19.1)
40–49 years	1011 (22.7)
50–59 years	1149 (25.8)
≥60 years	623 (14.0)
**Cardiovascular risk factors**	***n*** (%)
Yes	2592 (58.2)
**Cardiovascular disease**	***n*** (%)
Yes	125 (2.8)
**Highest educational level**	***n*** (%)
Lower secondary school or less	1114 (25.0)
High school	1897 (42.6)
University	1443 (32.4)

**Table 2 jcm-11-01695-t002:** Logistic regression for awareness of CVRF.

CVRF	Correctly Identified *n* (%)	Age (Years)				Level of Schooling		CVRF ^c^	CVD ^d^
		30–40 ^a^	40–50 ^a^	50–60 ^a^	≥60 ^a^	High School ^b^	University ^b^		
		OR (95%CI) *p*-Value	OR (95%CI) *p*-Value	OR (95%CI) *p*-Value	OR (95%CI) *p*-Value	OR (95%CI) *p*-Value	OR (95%CI) *p*-Value	OR (95%CI) *p*-Value	OR (95%CI) *p*-Value
Family CVD history	3804 (85.4)	1.02 (0.78–1.35) *p* = 0.844	1.12 (0.86–1.46) *p* = 0.416	0.95 (0.74–1.22) *p* = 0.668	1.51 (1.09–2.09) *p* = 0.014	1.64 (1.33–2.02) *p* < 0.0001	2.68 (2.09–3.43) *p* < 0.0001	1.31 (1.08–1.60) *p* = 0.007	1.11 (0.63–1.96) *p* = 0.725
Smoke	4343 (97.5)	1.34 (0.76–2.36) *p* = 0.312	1.20 (0.71–2.04) *p* = 0.491	3.09 (1.59–6.01) *p* = 0.001	1.54 (0.80–2.95) *p* = 0.194	1.75 (1.09–2.81) *p* = 0.021	1.72 (1.03–2.86) *p* = 0.037	1.44 (0.93–2.24) *p* = 0.107	2.77 (0.38–20.08) *p* = 0.313
High blood pressure	4316 (96.9)	1.42 (0.89–2.27) *p* = 0.141	1.88 (1.16–3.04) *p* = 0.010	3.72 (2.10–6.57) *p* < 0.0001	2.84 (1.48–5.43) *p* = 0.002	1.94 (1.31–2.88) *p* = 0.001	3.19 (1.94–5.24) *p* < 0.0001	1.32 (0.90–1.93) *p* = 0.150	3.58 (0.49–25.88) *p* = 0.206
High cholesterol	4271 (95.9)	0.97 (0.63–1.49) *p* = 0.875	1.24 (0.80–1.92) *p* = 0.335	2.34 (1.37–3.66) *p* = 0.001	1.72 (0.99–2.99) *p* = 0.055	1.92 (1.35–2.74) *p* < 0.0001	3.32 (2.12–5.19) *p* < 0.0001	1.70 (1.19–2.44) *p* = 0.004	2.16 (0.53–8.86) *p* = 0.283
Overweight	4360 (97.9)	1.62 (0.80–3.29) *p* = 0.178	1.47 (0.76–2.82) *p* = 0.251	1.57 (0.82–2.98) *p* = 0.172	0.88 (0.45–1.70) *p* = 0.884	2.23 (1.37–3.62) *p* = 0.001	3.94 (2.09–7.43) *p* < 0.0001	0.54 (0.52–1.40) *p* = 0.536	0.86 (0.52–1.40) *p* = 0.536
Physical inactivity	4258 (95.6)	1.55 (1.00–2.42) *p* = 0.050	1.86 (1.19–2.88) *p* = 0.006	1.83 (1.19–2.80) *p* = 0.005	1.44 (0.89–2.33) *p* = 0.133	1.44 (1.02–2.03) *p* = 0.039	2.35 (1.55–3.58) *p* < 0.0001	0.97 (0.69–1.37) *p* = 0.874	2.30 (0.56–9.43) *p* = 0.246
Diabetes	3910 (87.8)	1.51 (1.13–2.02) *p* = 0.005	1.12 (0.86–1.45) *p* = 0.400	1.61 (1.23–2.11) *p* = 0.001	2.15 (1.51–3.07) *p* < 0.0001	1.13 (0.89–1.42) *p* = 0.321	1.71 (1.31–2.24) *p* < 0.0001	0.97 (0.78–1.21) *p* = 0.776	1.68 (0.81–3.49) *p* = 0.162
Menopause	3692 (82.9)	0.87 (0.68–1.11) *p* = 0.266	0.91 (0.72–1.15) *p* = 0.437	1.99 (1.53–2.58) *p* < 0.0001	2.09 (1.52–2.87) *p* < 0.0001	1.39 (1.13–1.69) *p* = 0.002	1.96 (1.56–2.47) *p* < 0.0001	1.33 (1.11–1.61) *p* = 0.003	1.23 (0.71–2.14) *p* = 0.468

CVRF = cardiovascular risk factors; CVD = cardiovascular disease; OR = odds ratio. ^a^ Compared to 18–30 years old age group. ^b^ Compared to women with lower educational level (lower secondary school or less). ^c^ Compared to women without CVRF. ^d^ Compared to women without CVD.

**Table 3 jcm-11-01695-t003:** Characteristics of the three groups identified on the basis of the by SRH status.

	SRH Status			*p*-Value
	Poor	Acceptable	Good	
Age (years)	49.10 ± 15.16	46.54 ± 13.64	40.10 ± 12.93	<0.0001
CVRF %	64.4	65.4	47.7	<0.0001
CVD %	5.5	3.9	1.2	<0.0001

SRH = self-rated health; CVRF = cardiovascular risk factor; CVD = cardiovascular disease.

**Table 4 jcm-11-01695-t004:** Perceived CVD risk compared to demographics and clinical conditions.

	Perceived CVD Risk			*p*-Value
	Low	Intermediate	High	
Age (years)	41.27 ± 13.79	46.08 ± 13.49	50.14 ± 14.56	<0.0001
CVRF %	50.2	64.6	70	<0.0001
CVD %	1.7	2.9	10.2	<0.0001
SRH status %				<0.0001
Poor	9.9	9.8	19.5	
Acceptable	40	55.9	51.4	
Good	50.1	34.4	29.1	

CVD = cardiovascular disease; CVRF = cardiovascular risk factor; SRH = self-rated health.

## Data Availability

The data presented in this study are available on request from the corresponding author. The data are not publicly available due to privacy.

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
