# Peer review of "Cardiovascular Risk Perception and Knowledge among Italian Women: Lessons from IGENDA Protocol"

_jcm, 2022, doi:10.3390/jcm11061695_

Round 1
Reviewer 1 Report
This study is a true reflected of the degree of knowledge that the population has about cardiovascular risk factors and their association with gender, despite the years that they have been working on it.
It seems to me a job with an adequate methodology and with results that, although expected, always surprise. The authors adequately discuss their results. The only thing missing is much more concrete proposals/actions on the possibilities of improving the degree of knowledge and awareness of cardiovascular disease in the entire population.
Author Response
This study is a true reflected of the degree of knowledge that the population has about cardiovascular risk factors and their association with gender, despite the years that they have been working on it.
It seems to me a job with an adequate methodology and with results that, although expected, always surprise. The authors adequately discuss their results. The only thing missing is much more concrete proposals/actions on the possibilities of improving the degree of knowledge and awareness of cardiovascular disease in the entire population.
A: We would like to thank the Reviewer for his/her encouraging feedback. We strongly agree with his/her comment and the following sentences have been added. “This campaign should be conducted within the health system, during patient visits and by organizing specific seminars or conferences directed to the general population, and, above all, through various digital communication channels, including mass media (TV and radio advertisements and articles on newspapers and magazines), social media (blogs, micro-blogs, wikis, social networking sites, photo-sharing sites, instant messaging, video-sharing sites, podcasts), and text messages or phone calls. The digital communication campaign can reach audiences in a low-cost, impactful, and effective way.”.
Reviewer 2 Report
REVIEW FOR: Cardiovascular Risk Perception and Knowledge among Italian Women: Lessons from IGENDA Protocol
Thank you to the Editors for the opportunity to review this manuscript. I believe that this topic is important and relevant in the current context of women’s health. My main concerns are punctuation, grammar and referencing rigour.
- The Abstract features abbreviations such as CV, CVD CVRF – that seems too many for a short abstract and causes confusion. Maybe write the words out in the abstract and start using abbreviations in the main text as you also describe these terms and provide examples.
- You identify cardiovascular risk factors as CVRF, therefore you can use just CVRF rather than CVRFs as abbreviation throughout the paper
- Please write the abstract using the 3rd person.
- You may like to consider using rounded off %, rather than 69.8% (ie use 70%). Please apply throughout as it will make it easier to read. You may also like to consider using alternate terms to describe proportions here in the Abstract. In the case of the used example, consider saying ‘two-thirds’ or ‘the majority’. Too many percentages on top of abbreviations in a short abstract are distracting and do not aid the reader acclimatise to your manuscript
- There are some grammatical issues such as sentence structure. This therefore causes the message in the sentence to mean something different to what I presume the authors would like to say eg line 113 “The 60% of interviewed women….” - Please alter to: Sixty percent of the interviewed women considered CVD as an ……
Whilst the paper addresses overall knowledge of CVRF, it may be of importance to note that this is likely the best-case knowledge scenario given that other life events such as pregnancy related disorders (hypertensive disorders of pregnancy, gestational diabetes, preterm birth etc) may further compound the risk. Given your reference to the US and Australia, here is a relevant and recent paper you may like to include (I am not one of the authors) should you mention additional factors that increase CVRF further:
Arnott, C., Nelson, M., Alfaro Ramirez, M., Hyett, J., Gale, M., Henry, A., Celermajer, D. S., Taylor, L., & Woodward, M. (2020). Maternal cardiovascular risk after hypertensive disorder of pregnancy. Heart, 106(24), 1927-1933. https://doi.org/10.1136/heartjnl-2020-316541
And
Arnott, C., Patel, S., Hyett, J., Jennings, G., Woodward, M., & Celermajer, D. S. (2019). Women and Cardiovascular Disease: Pregnancy, the Forgotten Risk Factor. Heart, Lung and Circulation, 29(5), 662-667. https://doi.org/10.1016/j.hlc.2019.09.011
You will find further relevant references within the reference list of those papers.
- has your study identified whether the participants have had pregnancies and if so if they had any complications such as those listed above during one of their pregnancies?
- Line 132 – Italy being a lov CV risk country – needs a reference please.
- paragraph line 143 to 149 you refer to US and Australia, then the examples refer to the US and Austria… is there a mistake? The reference refers to Austrian data, so I presume it is intentional. This paragraph jumps a little and is not clear enough.
- Avoid starting a sentence with ‘Accordingly’ it does not read well.
- Avoid sentences such as line 146 – ‘The survey….” When the survey has not been referred to prior to this sentence. You need to provide more context
- line 151 instead of ‘even in women’ please change to ‘even amongst women…’
- line 156 – you refer to 90% of the published studies – please reference them here.
- There are a number of paragraphs when really, they are just sentences. These can be fused into more appropriate paragraphs. Please review grammar, sentence structure, paragraph structure and overall improved referencing please.
- line 166 this statement and reference to a national health survey lacks a reference.
- line 176 – anonymous, self-administered (you need a comma, you are listing adjectives)
- please reference the Declaration of Helsinki.
- Line 187 – do not start a sentence or a section with ‘In short”
- Line 188 – it should say, “…the following, three main parts: ….”
- The layout of the Method esp under 2.2 is confusing. Why are there separate paragraphs for each sentence at the end? This should all be written in one paragraph. The same applies to the beginning on 2.3
- line 207 there are 2 spaces between a full-stop and the next word.
- Table 1 should also feature numbers, not just %
- Fig 1 a and b – in one title the authors use ‘CVD are the main…” in the other title “ CVD is an almost…’ please review the grammar here.
- Please review the rules for the use of the apostrophe throughout.
Table 2 - please include numbers and % in the column correctly identified.
- please write p-value the same way throughout. You use at least two different ways (p-value and P-value – choose one and use throughout)
You feature Figure 1 twice – I believe the second is Figure 2?
Table 3 – possibly change the word ‘bad’ to ‘poor’
In Table 3 you refer to the p value as p value – again, pick one way to write it and refer to this same way throughout.
- I do not think SRH is defines in the text- just under a table. I would suggest to write this one out throughout. Too many abbreviations and it becomes confusing
- Line 293 – What makes this sample a ‘representative’ sample? With an Italian population og 60.5 million, of which 51% are female, 4454 women does not necessarily seem representative.
- Please condense your findings and avoid repetition at the beginning of your Discussion section.
- line 319 – please review punctuation and grammar in this sentence
- Being overweight (rather then just overweight) was the most… (line 320)
- I suggest the discussion focus a little more on the critical analysis of existing studies with quick reference to the author’s own study, rather than reiterating the results in quite as much detail.
- When referring to an author in a sentence, please reference immediately after the name rather than at the end of the sentence. Eg line 335 – ‘Hoare et al. (15) revealed….’
- line 337 – It has consistently been shown…. (rather than starting a sentence with Consistently, …)
- line 353 – lack of awareness rather than unawareness
- Line 355 – A Canadian study… (not The Canadian study – ‘the’ is too specific)
Some items in the discussion are a little confusing or too specific and they distract from the overall message within the discussion
Although the birth rate in Italy is low (approx. 1.2 births per woman), I would like to see the discussion of CVD risk refer to the context of women experiencing normal life events (such pregnancy) and how complications during these normal life events can further compound the already existing CVRF. The overall ‘so what’ of this study has great potential here in the discussion.
Author Response
We would like to thank the Editor and the Reviewer for their encouraging feedback and constructive critique and for the effort regarding this manuscript. We have addressed each of the raised concerns, which have substantially improved the manuscript.
Thank you to the Editors for the opportunity to review this manuscript. I believe that this topic is important and relevant in the current context of women’s health. My main concerns are punctuation, grammar and referencing rigour.
- The Abstract features abbreviations such as CV, CVD CVRF – that seems too many for a short abstract and causes confusion. Maybe write the words out in the abstract and start using abbreviations in the main text as you also describe these terms and provide examples.
A: Thank you for this comment. We have now left only one abbreviation: CVD.
- You identify cardiovascular risk factors as CVRF, therefore you can use just CVRF rather than CVRFs as abbreviation throughout the paper
A: Following thee Reviewer’s suggestion, we have now used as abbreviation only CVRF rather than CVRFs.
- Please write the abstract using the 3rd person.
A: The 3rs person has now been used in the Abstract.
- You may like to consider using rounded off %, rather than 69.8% (ie use 70%). Please apply throughout as it will make it easier to read. You may also like to consider using alternate terms to describe proportions here in the Abstract. In the case of the used example, consider saying ‘two-thirds’ or ‘the majority’. Too many percentages on top of abbreviations in a short abstract are distracting and do not aid the reader acclimatise to your manuscript
A: Thank you for these suggestion. We have now used 70% instead of 69.8%.
- There are some grammatical issues such as sentence structure. This therefore causes the message in the sentence to mean something different to what I presume the authors would like to say eg line 113 “The 60% of interviewed women….” - Please alter to: Sixty percent of the interviewed women considered CVD as an ……
A: We have done the suggested correction.
Whilst the paper addresses overall knowledge of CVRF, it may be of importance to note that this is likely the best-case knowledge scenario given that other life events such as pregnancy related disorders (hypertensive disorders of pregnancy, gestational diabetes, preterm birth etc) may further compound the risk. Given your reference to the US and Australia, here is a relevant and recent paper you may like to include (I am not one of the authors) should you mention additional factors that increase CVRF further:
Arnott, C., Nelson, M., Alfaro Ramirez, M., Hyett, J., Gale, M., Henry, A., Celermajer, D. S., Taylor, L., & Woodward, M. (2020). Maternal cardiovascular risk after hypertensive disorder of pregnancy. Heart, 106(24), 1927-1933. https://doi.org/10.1136/heartjnl-2020-316541
And
Arnott, C., Patel, S., Hyett, J., Jennings, G., Woodward, M., & Celermajer, D. S. (2019). Women and Cardiovascular Disease: Pregnancy, the Forgotten Risk Factor. Heart, Lung and Circulation, 29(5), 662-667. https://doi.org/10.1016/j.hlc.2019.09.011
You will find further relevant references within the reference list of those papers.
A: Thank you for this comment. The following sentence (with references) has now been added in the text. “Moreover, several female-specific risk factors for CVD, such as polycystic ovary syn-drome, premature ovarian failure, hypertensive disorders of pregnancy, gestational diabetes, and preterm birth, are under appreciated clinically and not receive adequate attention in CVD risk prediction (ref).”
- has your study identified whether the participants have had pregnancies and if so if they had any complications such as those listed above during one of their pregnancies?
A: Unfortunately, this information has not been verified.
- Line 132 – Italy being a lov CV risk country – needs a reference please.
A: The following reference has been added: Conroy RM, Pyorala K, Fitzgerald AP, et al. Estimation of ten-year risk of fatal cardiovascular disease in Europe: the SCORE project. Eur Heart J. 2003;24(11):987-1003.
- paragraph line 143 to 149 you refer to US and Australia, then the examples refer to the US and Austria… is there a mistake? The reference refers to Austrian data, so I presume it is intentional. This paragraph jumps a little and is not clear enough.
A: The knowledge of CVD as the leading cause of death was suboptimal among Australian and US women but good among Austrian women.
- Avoid starting a sentence with ‘Accordingly’ it does not read well.
A: “Accordingly” has been eliminated.
- Avoid sentences such as line 146 – ‘The survey….” When the survey has not been referred to prior to this sentence. You need to provide more context
A: The sentence has been modified as follows. “A nationwide survey conducted by the Women's Heart Alliance in 2014 to determine barriers and opportunities for women and physicians with regard to CVD showed that almost half of 1011 interviewed women did not recognize CVD as the number one killer of women in the US (ref)”.
- line 151 instead of ‘even in women’ please change to ‘even amongst women…’
A: We have done the suggested correction.
- line 156 – you refer to 90% of the published studies – please reference them here.
A: References have been added.
- There are a number of paragraphs when really, they are just sentences. These can be fused into more appropriate paragraphs. Please review grammar, sentence structure, paragraph structure and overall improved referencing please.
A: We have followed this Reviewer’s suggestion.
- line 166 this statement and reference to a national health survey lacks a reference.
A: We have added the following reference: Maffei S, Cugusi L, Meloni A, et al. IGENDA protocol: gender differences in awareness, knowledge and perception of cardiovascular risk: an Italian multicenter study. J Cardiovasc Med (Hagerstown). 2019;20(5):278-283.
- line 176 – anonymous, self-administered (you need a comma, you are listing adjectives)
A: A comma has been added.
- please reference the Declaration of Helsinki.
A: The current (2013) version has been cited.
- Line 187 – do not start a sentence or a section with ‘In short”
A: “In short” has now been deleted.
- Line 188 – it should say, “…the following, three main parts: ….”
A: The comma has been added.
- The layout of the Method esp under 2.2 is confusing. Why are there separate paragraphs for each sentence at the end? This should all be written in one paragraph. The same applies to the beginning on 2.3
A: The two first paragraphs of 2.3 sections have been unified.
- line 207 there are 2 spaces between a full-stop and the next word.
A: One space has been deleted.
- Table 1 should also feature numbers, not just %
A: Numbers have been added in Table 1.
- Fig 1 a and b – in one title the authors use ‘CVD are the main…” in the other title “ CVD is an almost…’ please review the grammar here.
A: Figure 1 has been modified: “CVD is” has been used.
- Please review the rules for the use of the apostrophe throughout.
A: Thank you for this suggestion.
Table 2 - please include numbers and % in the column correctly identified.
A: Numbers have been included in Table 2.
- please write p-value the same way throughout. You use at least two different ways (p-value and P-value – choose one and use throughout)
A: We have now used P-value throughout the text.
You feature Figure 1 twice – I believe the second is Figure 2?
A: The Figure caption has been corrected.
Table 3 – possibly change the word ‘bad’ to ‘poor’
A: We have done the required change.
In Table 3 you refer to the p value as p value – again, pick one way to write it and refer to this same way throughout.
A: We have now used P-value throughout the text.
- I do not think SRH is defines in the text- just under a table. I would suggest to write this one out throughout. Too many abbreviations and it becomes confusing
A: SRH is defined in the Methods section.
- Line 293 – What makes this sample a ‘representative’ sample? With an Italian population og 60.5 million, of which 51% are female, 4454 women does not necessarily seem representative.
A: We have now deleted the term “representative”.
- Please condense your findings and avoid repetition at the beginning of your Discussion section.
A: The fist, repetitive sentence of the Discussion has been replaced by the following short sentence: “ This national survey documented several important findings”.
- line 319 – please review punctuation and grammar in this sentence
A: We have reviewed punctuation and grammar.
- Being overweight (rather then just overweight) was the most… (line 320)
A: We have done the suggested correction.
- I suggest the discussion focus a little more on the critical analysis of existing studies with quick reference to the author’s own study, rather than reiterating the results in quite as much detail.
A: We thank the Reviewer for this suggestion, that we have tried to follow.
- When referring to an author in a sentence, please reference immediately after the name rather than at the end of the sentence. Eg line 335 – ‘Hoare et al. (15) revealed….’
A: We have done the suggested correction.
- line 337 – It has consistently been shown…. (rather than starting a sentence with Consistently, …)
A: We have done the suggested correction.
- line 353 – lack of awareness rather than unawareness
A: We have done the suggested correction.
- Line 355 – A Canadian study… (not The Canadian study – ‘the’ is too specific)
A: We have done the suggested correction.
Some items in the discussion are a little confusing or too specific and they distract from the overall message within the discussion
A: we have tried to make the Discussion more fluid.
Although the birth rate in Italy is low (approx. 1.2 births per woman), I would like to see the discussion of CVD risk refer to the context of women experiencing normal life events (such pregnancy) and how complications during these normal life events can further compound the already existing CVRF. The overall ‘so what’ of this study has great potential here in the discussion.
A: We have investigated only the knowledge of the traditional risk factors, without taking into account female-specific factors. The following sentences have been added in the Discussiom. “Besides menopause, the knowledge of women-specific CVRF was not evaluated, but we are planning to explore this issue in our population. Two studies conducted in the USA showed that women with pre-eclampsia or gestational diabetes mellitus were unfamiliar with the relationship between these conditions and increased future CVD risk (ref).”